# Real-World Clinical Outcomes of Molnupiravir for the Treatment of Mild to Moderate COVID-19 in Adult Patients during the Dominance of the Omicron Variant: A Meta-Analysis

**DOI:** 10.3390/antibiotics12020393

**Published:** 2023-02-15

**Authors:** Chienhsiu Huang, Tsung-Lung Lu, Lichen Lin

**Affiliations:** 1Department of Internal Medicine, Dalin Tzu Chi Hospital, Buddhist Tzu Chi Medical Foundation, NO. 2, Min-Sheng Road, Dalin Town, Chiayi 62247, Taiwan; 2Department of Nursing, Dalin Tzu Chi Hospital, Buddhist Tzu Chi Medical Foundation, NO. 2, Min-Sheng Road, Dalin Town, Chiayi 62247, Taiwan

**Keywords:** COVID-19, molnupiravir, all-cause mortality, composite outcome of disease progression, COVID-19 vaccination coverage, viral load

## Abstract

Introduction: The therapeutic impact of molnupiravir in the Omicron variant phase is unknown. The goal of the current meta-analysis was to compare the real-world clinical outcomes of molnupiravir for the treatment of mild to moderate COVID-19 during the dominance of the Omicron variant in adult patients to that of a placebo. Methods: To be included, studies had to directly compare the clinical effectiveness of molnupiravir in treating adult COVID-19 patients to that of a placebo. Studies were included based on the following outcomes: all-cause mortality, composite outcome of disease progression, hospitalization rate, and viral load. Results: The current meta-analysis included six studies that indicated that the risk of mortality was reduced by 34%, and the risk of composite outcome of disease progression was reduced by 37% among patients who received molnupiravir. Molnupiravir was associated with faster reduction in viral loads than the placebo. There was no clinical benefit of reducing all-cause mortality in mild to moderate COVID-19 patients with high COVID-19 vaccination coverage. Conclusion: The clinical effectiveness of molnupiravir was associated with COVID-19 vaccination coverage in COVID-19 patients. There is a lack of detailed data on its effectiveness in vaccinated patients, especially those with low COVID-19 vaccination coverage.

## 1. Introduction

The coronavirus disease 2019 (COVID-19) pandemic has resulted in a high rate of patient mortality and exerted a considerable impact on human activities and the global economy. Clinically, most patients with severe acute respiratory syndrome coronavirus 2 (SARS-CoV-2) infection experience no symptoms or develop mild illness requiring outpatient treatment, However, a small portion develop life-threatening disease and respiratory failure [1,2]. Therefore, preventing SARS-CoV-2 infection from developing into severe COVID-19 is an important issue. Vaccines and antiviral drugs are two important treatment modalities for the prevention and treatment of COVID-19. Despite the possibility that the most effective method for controlling COVID-19 is vaccination, widespread public immunization will take a long time due to public acceptance issues regarding vaccination. In addition, reports of vaccinated adults who contract SARS-CoV-2 infection has been increasing worldwide. Therefore, oral antiviral therapy is urgently needed, as effective antiviral therapy can prevent disease progression and stop the spread of the virus. To address this, two new oral antiviral drugs that can be administered at home by patients shortly after diagnosis have recently been developed. Molnupiravir and nirmatrelvir plus ritonavir have received emergency use authorization from the Food and Drug Administration (FDA) of the United States for the treatment of mild to moderate COVID-19 in adult patients who are not hospitalized and have a high risk of developing severe disease [3,4]. Nirmatrelvir 300 mg and ritonavir 100 mg are the suggested doses for this medication. Based on the results of the EPIC-HR study, it was found that nirmatrelvir and ritonavir significantly decreased the risk of hospitalization or death due to COVID-19 by 89% among unvaccinated patients within 3 days of symptom onset and by 88% among unvaccinated patients within 5 days of symptom onset [5].

Molnupiravir is a beta-d-N4-hydroxycytidine (NHC) ribonucleoside prodrug that inhibits SARS-CoV-2 ribonucleic acid (RNA)-dependent RNA polymerase. Molnupiravir triphosphate (MTP), the active form of NHC that is incorporated into viral RNA by SARS-CoV-2 RNA polymerase after oral administration, is phosphorylated intracellularly and converted in the cell. It misdirects the viral polymerase to incorporate either adenosine or guanosine during viral replication after incorporation, rendering the virus ineffective and resulting in fatal mutations [6,7,8]. Molnupiravir is a small-molecule ribonucleoside that can be taken orally as an NHC prodrug. Molnupiravir has strong activity against coronaviruses, including SARS-CoV-2 variants, and a high barrier against the development of drug resistance [9,10,11]. For adults over the age of 18, the current recommended dosage is 800 mg (four 200-mg capsules) taken orally every 12 h for five days. For COVID-19 patients, treatment with molnupiravir should begin within five days of the onset of symptoms. The likelihood of drug interactions is limited because molnupiravir is a prodrug that is metabolized by human esterases into its active form. Molnupiravir can easily be administered at home and has no interaction with other chronic treatments [12]. Molnupiravir is currently not recommended for use in children or pregnant individuals, as its safety and effectiveness have not been established, and people who have childbearing potential should use effective contraception. Molnupiravir exposure through breastfeeding may also cause adverse effects in infants [13]. Minotti C et al. reported the clinical off-label prescription experience of two children with COVID-19 who received molnupiravir [14]. Molnupiravir is a therapy that can be used instead of Paxlovid when this is not available, not possible to use, or not clinically appropriate [1].

According to the COVID-19 treatment guidelines published by the National Institutes of Health, molnupiravir showed a 31% lower rate of hospitalization or death than placebo in the MOVe-OUT study [15]. A number of studies have demonstrated that the SARS-CoV-2 Omicron variant shows high transmissibility, partial vaccine escape, and a lesser clinical severity of illness than the Delta variant [16,17,18,19,20]. In 2021, prior to the advent of Omicron, data from the “MOVe-OUT” experiment were collected in unvaccinated patients. The therapeutic impact of molnupiravir at this time compared to the less virulent Omicron versions and in groups that have previously received vaccinations were not addressed by these findings. It is unknown if the stated advantages will continue in a population that has received vaccinations. At present, clinical studies on molnupiravir are ongoing. We conducted this meta-analysis to evaluate the real-world clinical benefit among COVID-19 patients who received molnupiravir therapy during the dominance of the Omicron variant and aimed to assess the effectiveness of molnupiravir for the treatment of mild to moderate COVID-19 in adult patients, irrespective of vaccination status.

## 2. Method

The meta-analysis followed the reporting of the PRISMA guidelines, and a completed PRISMA flow diagram is included as part of the main text.

### 2.1. Data Search and Extraction

Between 1 January 2020 and 31 December 2022, a comprehensive literature search in the PubMed, Web of Science, and Cochrane Library databases yielded all clinical studies. We searched for “Molnupiravir,” “COVID-19,” “coronavirus 2019,” and “SARS-CoV-2.” Studies that directly compared the clinical effectiveness of molnupiravir to a placebo in the treatment of adult COVID-19 patients were considered eligible for inclusion. Retrospective and prospective observational studies, as well as randomized controlled trials (RCTs), made up the relevant clinical studies. To eliminate duplicate records, articles in each database that contained relevant terms were identified and imported into EndNote. After excluding duplicates, each study’s title and/or abstract were read to identify irrelevant studies. Two authors independently reviewed the titles and abstracts of the identified trial reports to determine their eligibility. All potentially relevant articles were reviewed by reading the full texts to identify eligible trial reports after excluding irrelevant studies. Data were manually extracted from eligible full-text articles. Information on authors, region, study design, study period, SAR-CoV-2 variant of concern, severity of COVID-19, timing of COVID-19 infection, mean age, vaccination history, total number of molnupiravir-treated patients, total number of placebo-treated patients, dosage of molnupiravir, hospitalization rate, all-cause mortality, and composite outcome of disease progression was extracted.

### 2.2. Inclusion Criteria

Because there were only a few RCTs available, we compensated for the limitations of the data analysis by including the outcomes of observational studies. The studies were considered eligible for inclusion only if they directly compared the clinical effectiveness of molnupiravir versus a placebo in the treatment of COVID-19 in adult patients. Studies that had any one or more of the following outcomes were included: rate of hospitalization, mortality from all causes, and composite outcome of disease progression. Patients received doses of molnupiravir 800 mg every 12 h for 5 days, and treatment should begin within five days of the onset of signs or symptoms.

### 2.3. Exclusion Criteria

Our analysis did not include unpublished articles or treatment regimens that did not include molnupiravir. The exclusion criteria included individuals under the age of 18. The meta-analysis explored the real-world clinical outcome of molnupiravir for the treatment of mild to moderate COVID-19 in adult patients during the dominance of the Omicron variant. We excluded studies during the non-Omicron variant phase and clinical trials in the current meta-analysis. There are four Hong Kong studies in the literature. All four studies were carried out from February 2022 to June 2022 and contained the same patients. Therefore, we excluded two studies in the current meta-analysis [21,22].

### 2.4. Definitions

All-cause mortality was defined as death from any cause in COVID-19 patients. Fully vaccinated patients were defined as those with at least two doses of the COVID-19 vaccine. High COVID-19 vaccination coverage was defined as more than 70% fully vaccinated patients. Low COVID-19 vaccination coverage defined fully vaccinated patients as less than 71%. Composite disease progression outcomes included hospitalization, all-cause mortality, initiation of invasive mechanical ventilation, intensive care unit admission, and need for oxygen therapy.

### 2.5. Quality Assessment

We assessed the risk of bias in each study using the Cochrane Risk-of-Bias Tool 2.0 for RCTs. The Risk of Bias in Non-randomized Studies of Interventions (ROBINS-I) tool was used to evaluate observational studies [23]. We conducted a sensitivity analysis by systematically removing each study and assessed the impact of the study quality on the effect estimates. Quality of the evidence was ranked based on the risk of bias according to the Grading of Recommendations Assessment, Development and Evaluation (GRADE) approach at the outcome level [24,25]. Two reviewers examined publications independently to avoid bias. When disagreement occurred, a third author resolved the issue.

### 2.6. Statistical Analysis

Data were entered into the Cochrane Review Manager software RevMan 5. Differences were expressed as odds ratios (ORs) with 95% confidence intervals (CIs) for dichotomous outcomes. The significance of the pooled ratios was determined by the Z test, and a *p* value less than 0.05 was considered statistically significant. The I^2^ test was used to assess the proportion of statistical heterogeneity, and the Q-statistic test was used to define the degree of heterogeneity. A *p* value less than 0.10 for the Q-test and I^2^ less than 50% were considered significant among the studies. The fixed-effects model was used when the effects were assumed to be homogenous, while the random-effects model was used when they were heterogeneous. Publication bias was assessed by examining the funnel plot.

## 3. Results

### 3.1. Characteristics of the Included Studies

The specifics of the procedure for selecting a study are depicted in the flowchart in Figure 1.

From the PubMed, Web of Science, and Cochrane Library databases, there were 239, 252, and 39 initial search results, respectively. There were 134 duplicate articles. Reading the title or abstract revealed a total of 396 irrelevant studies. There were 56 potentially relevant articles left after removing duplicates and irrelevant studies. Fifty articles were omitted from a full-text review, including the absence of molnupiravir versus placebo results for adult COVID-19 patients, unpublished articles, clinical trials, and the two studies of Hong Kong. Six studies were included in the final meta-analysis [26,27,28,29,30,31]. Table 1 displays the primary characteristics of the six included studies. Five of the studies had a high risk of bias [27,28,29,30,31]. Figure 2 and Table 2 show the assessment of the bias risk.

### 3.2. Effectiveness and Safety Outcomes 

Six studies involving 89,480 patients (22,355 who received molnupiravir therapy, 67,125 who received placebo therapy) reported all-cause mortality [26,27,28,29,30,31]. All-cause mortality was significantly different between molnupiravir-treated and placebo-treated patients (RR = 0.66, 95% CI = 0.49–0.88, *p* = 0.004, I^2^ = 54%) (Figure 3).

Three studies involving 31,296 patients with high COVID-19 vaccination coverage (15,420 who received molnupiravir therapy, 15,876 who received placebo therapy) reported all-cause mortality [26,28,29]. All-cause mortality was not significantly different between molnupiravir-treated and placebo-treated patients (RR = 0, 95% CI = 0.00–0.00, *p* = 0.48, I^2^ = 0%) (Figure 4).

Two studies involving 57,929 patients with low COVID-19 vaccination coverage (6839 who received molnupiravir therapy, 51,090 who received placebo therapy) reported all-cause mortality [30,31]. All-cause mortality was significantly different between molnupiravir-treated and placebo-treated patients (RR = 0.62, 95% CI = 0.42–0.93, *p* = 0.02, I^2^ = 88%) (Figure 5).

Four studies involving 64,171 patients (9730 who received molnupiravir therapy, 54,441 who received placebo therapy) reported the composite outcome of disease progression [28,29,30,31]. The composite outcome of disease progression was significantly different between molnupiravir-treated and placebo-treated patients (RR = 0.63, 95% CI = 0.57–0.71, *p* < 0.001, I^2^ = 10%) (Figure 6).

Two studies involving 79,271 patients (17,512 who received molnupiravir therapy, 61,759 who received placebo therapy) reported hospitalization [26,31]. Hospitalization was not significantly different between molnupiravir-treated and placebo-treated patients (RR = 0.98, 95% CI = 0.90–1.07, *p* = 0.66, I^2^ = 0%) (Figure 7).

## 4. Discussion

Adult patients with mild to moderate COVID-19 who were treated with molnupiravir had a lower risk of all-cause mortality and composite outcome of disease progression than those treated with a placebo, according to the current meta-analysis of six studies. The risk of all-cause mortality was reduced by 34%, and the risk of the composite outcome of disease progression was reduced by 37% among patients who received molnupiravir. Molnupiravir was associated with a similar hospitalization rate as placebo.

A brief meta-analysis (letter to the editor) examined the effectiveness of molnupiravir in treating adults with COVID-19. Six RCTs were included in the meta-analysis, and the results revealed that molnupiravir had no benefit for hospitalized patients and showed a decreased mortality risk for non-hospitalized patients when compared to placebo [32]. This brief meta-analysis included two unpublished articles [33,34]. In addition, the study of Caraco Y et al. is the interim analysis of the full MOVe-out study by Bernal, and the data are duplicated [35]. The author cited only one RCT related to COVID-19 in hospitalized patients [36]. The study by Arribas JR et al. included 99 (43.8%) severe COVID-19 patients in the molnupiravir therapy group. In addition, 175 (77.4%) COVID-19 patients had an interval from signs and symptoms to treatment of more than 5 days. Arribas concluded that molnupiravir therapy for the treatment of COVID-19 did not demonstrate clinical benefit [36]. As part of the National Institutes of Health (NIH) COVID-19 treatment guidelines, adult patients who have not been admitted to a hospital and those who have a high risk of disease progression should receive 800 mg of molnupiravir orally twice daily for five days. Treatment should begin as soon as possible within five days of the onset of signs or symptoms [1]. In the Arribas JR et al. study, there were many severe COVID-19 patients, and the molnupiravir therapy group started treatment more than five days after signs and symptoms started [35]. Therefore, the study found no clinical benefit of molnupiravir therapy for COVID-19 patients. In the current meta-analysis, the risk of mortality was reduced by 34% among adult patients who received molnupiravir for the treatment of COVID-19. We performed subgroup analysis and found the benefit of reducing all-cause mortality in patients with low COVID-19 vaccination coverage only [30,31]. There was no benefit of reducing all-cause mortality in patients with high COVID-19 vaccination coverage [26,28,29]. It seems that a population with high COVID-19 vaccination coverage is a very important factor in reducing all-cause mortality for COVID-19 patients.

Regarding the composite outcome of disease progression in the current meta-analysis, four studies reported the composite outcome of disease progression [28,29,30,31]. All four studies showed that adults who took molnupiravir to treat COVID-19 had a lower risk of composite outcome of disease progression. Najjar-Debbiny R et al. highlighted that patients with inadequate COVID-19 vaccination and older age had a significantly lower risk of the composite outcome of disease progression [28]. Bruno G et al. analyzed SARS-CoV-2 infections in 554 patients (93.5% COVID-19 vaccination coverage, mean age 73.0 years) who had been fully vaccinated and receiving molnupiravir therapy. Forty-seven patients (8.5%) were associated with composite outcomes, including thirty-six all-cause hospitalizations and eleven deaths. One hundred sixty-five patients (90.9% COVID-19 vaccination coverage, mean age 62.0 years) had been fully vaccinated and received nirmatrelvir plus ritonavir therapy. Nine patients (5.5%) were associated with composite outcomes, including seven all-cause hospitalizations and two deaths. This study showed that early administration of both oral antivirals reduced composite outcomes among older COVID-19 outpatients with high COVID-19 vaccination coverage [37]. Tiseo G et al. analyzed mild-to-moderate SARS-CoV-2 infections in 114 patients (median age: 69.5 years) who had been adequately vaccinated (74.6% COVID-19 vaccination coverage) and were receiving molnupiravir therapy in their study. Two patients (1.8%) had composite outcomes, including one all-cause hospitalization and one death. Two hundred fifty-two patients (median age: 65 years) had been adequately vaccinated (86.9% COVID-19 vaccination coverage) and received nirmatrelvir plus ritonavir therapy. Two patients (0.8%) experienced composite outcomes, including one all-cause hospitalization and one death. The authors concluded that in the real-world non-hospitalization management of COVID-19 patients, the risk of composite outcomes of disease progression (hospitalization or death) did not differ between the oral antivirals [38]. The study by Tiseo G et al. showed that high COVID-19 vaccination coverage among older patients treated with antivirals could reduce all-cause hospitalizations and deaths. 

Regarding the hospitalization rate, two studies reported the hospitalization rate in the current meta-analysis and demonstrated that molnupiravir was associated with a similar hospitalization rate as placebo in non-hospitalized COVID-19 patients [26,31]. Only two included studies could not make definite conclusions about the hospitalization rate of COVID-19 outpatients in the current meta-analysis. In addition, the criteria for admission hospitalization at each hospital were heterogeneous. Further RCTs should be conducted to evaluate the benefit of molnupiravir related to the hospitalization rate of non-hospitalized COVID-19 patients.

Regarding the reduction in viral load, three studies reported that adult patients who received molnupiravir therapy for the treatment of COVID-19 had significantly lower viral load than those who received a placebo in the literature [15,34,39]. In the current meta-analysis, the study of Butler CC et al. showed that the viral load was significantly lower in patients receiving molnupiravir than in placebo patients at day 5 [26]. The study of Wong CK et al. showed that the time to achieve a low viral load (Ct ≥ 30) was significantly shortened for molnupiravir patients compared to placebo patients [30]. We conclude that oral molnupiravir for COVID-19 patients can reduce viral loads faster than placebo. The benefit of faster reduction in viral loads of COVID-19 patients can decrease SARS-CoV-2 transmission in the public. Decreasing SARS-CoV-2 transmission in the public is an important policy to control COVID-19 in the real world.

Regarding adverse events, the common adverse events among molnupiravir treatment patients were diarrhea, nausea, dysgeusia, headache and dizziness. Skin rash and urticaria were uncommon. All observed adverse events were mild and did not lead to drug discontinuation [40,41,42,43,44,45]. The meta-analysis by Amani B et al. explored the potential side effects of molnupiravir in COVID-19 patients. The authors claimed that there was no statistically significant difference between the molnupiravir and placebo groups in terms of adverse events. There was no significant correlation between adverse events that resulted in death or treatment discontinuation. In patients with COVID-19, molnupiravir was found to be a safe treatment option that was well tolerated [46]. In the study by Mazzitelli M et al., 28/407 (6.9%) patients experienced any adverse events following oral molnupiravir: 18 (4.4%) patients who reported adverse events presented grade 1 adverse events, 8 (1.9%) presented grade 2 adverse events and 2 (0.4%) presented grade 3 adverse events. The most reported side effects were nausea/vomiting (2%), bloating (1.7%), and dysgeusia (1.2%). Two patients (0.5%) reported a severe hypersensitivity reaction [47].

In our current meta-analysis, the study of Butler CC et al. also showed that molnupiravir was a safe oral antiviral agent and was well tolerated in adult COVID-19 patients [26].

In a study by Vena A. et al., fully vaccinated adult patients (high COVID-19 vaccination coverage, mean age 71.0 years) with mild to moderate COVID-19 received molnupiravir therapy. According to the study, 2.7% (4/145) of the patients required hospitalization; however, none of them passed away or were admitted to the intensive care unit (ICU) during the 30-day follow-up. The results of Vena’s study showed a much better clinical outcome for COVID-19-vaccinated patients who received molnupiravir [48]. Arbel R et al. (unpublished study) analyzed real-world data of 19,868 non-hospitalized COVID-19 patients (median age: 69.1 years) with prior COVID-19 vaccination during the dominant period of the Omicron variant. The study compared the clinical effectiveness of molnupiravir to a placebo in the treatment of adult COVID-19 patients. Among patients 65 years of age and older, 4 of 845 patients treated with molnupiravir and 137 of 12,724 untreated patients died due to COVID-19. A total of 18 of 845 patients treated with molnupiravir and 513 of 12,724 untreated patients aged 65 and older were hospitalized for COVID-19. The authors concluded that individuals 65 years of age and older who received molnupiravir had significantly reduced numbers of hospitalizations and deaths. However, there was no evidence of a benefit for younger adults [49]. Gentile I et al. found that 56.8% of 257 patients received molnupiravir, whereas 43.2% received nirmatrelvir/ritonavir. There were three hospitalizations in the molnupiravir (2.1%) group and one in the nirmatrelvir/ritonavir (0.9%) group. One patient who received molnupiravir died. The authors concluded that in a cohort of mostly vaccinated individuals treated with oral antivirals, hospitalization and death rates were low [50]. The study results were not surprising in either group, as 247 (96.1%) individuals had received at least two doses of a COVID-19 vaccine. In fact, SARS-CoV-2 vaccination likely influenced the overall outcome observed in the study population. In the PANORAMIC study, more than 90% of patients were fully vaccinated, and the study concluded that molnupiravir did not reduce hospitalizations or all-cause mortality among COVID-19 patients [26]. Medical experts offered their opinions on this study [51,52,53]. Molnupiravir did not reduce hospitalizations or all-cause mortality among those who had been vaccinated, but there may be other benefits to the therapy, such as a faster recovery and less follow-up with medical services [51]. The PANORAMIC study’s findings provided an intriguing combination of hopeful benefits associated with early recovery and likely reduced spread, but molnupiravir did not lower the frequency of hospitalizations or deaths related to COVID-19 among high-risk vaccinated adults in the community. The full clinical impact and cost-effectiveness of molnupiravir will not be known until all planned analyses are completed and a better understanding of the long-term evolution of variants is gained [52]. In PANORAMIC, the primary outcome was all-cause hospitalization or death. The population in the study was young (mean age, 56.6 years). Individuals 75 years or older made up just 1638 (approximately 6%) of the total number. As a result, the study group was not representative of the UK population that was most likely to experience hospitalization and death due to COVID-19. The study results were not surprising, especially the low rate of hospitalization or death in both groups, as 24,290 (94%) individuals had received at least three doses of a COVID-19 vaccine [53]. In the real world, few countries can achieve a 90% full vaccination coverage of the population, especially less developed countries. In the current meta-analysis, the mean age of participants was less than 60 years only in the Butler CC et al. study [26]. The study by Wong CK et al. included 88.7 participants over the age of 60 [31]. The studies by Najjar-Debbiny R et al. and Arbel R et al. found that older COVID-19 patients who received molnupiravir had better clinical outcomes than younger adults [28.49]. The clinical effectiveness of molnupiravir was also related to the age of COVID-19 patients. There are numerous challenges associated with using molnupiravir to treat COVID-19 in adult patients, including a lack of detailed data on its effectiveness in vaccinated patients, especially those with low COVID-19 vaccination coverage. To better comprehend the effectiveness of molnupiravir in the treatment of mild to moderate COVID-19 in adult patients, the medical community must immediately conduct more RCTs. This information should be taken into account by doctors when making decisions about how to treat adults with mild to moderate COVID-19.

## 5. Limitations

This topic has been the subject of only one RCT. The results of observational studies were incorporated into the current meta-analysis. Neither selection bias nor confounding could be eliminated. In addition, the six studies included in the current meta-analysis showed heterogeneity due to different populations and different vaccine histories. As a result, the quality of the six studies varied widely, and the results varied as well. This is another limitation of the current meta-analysis.

## 6. Conclusions

The current meta-analysis of the real-world Omicron variant phase found that the clinical effectiveness of molnupiravir was associated with COVID-19 vaccination coverage and age in COVID-19 patients. Molnupiravir showed clinical effectiveness in all-cause mortality and composite outcome of disease progression for low COVID-19 vaccination coverage in adult patients with mild to moderate COVID-19 compared with placebo. Molnupiravir showed clinical effectiveness in the composite outcome of disease progression for high COVID-19 vaccination coverage in adult patients with mild to moderate COVID-19 compared with placebo. There was no clinical benefit of reducing all causes of mortality in patients with high COVID-19 vaccination coverage. Molnupiravir was associated with faster decline in viral loads than placebo. Molnupiravir was not inferior to the placebo in terms of safety.

## Figures and Tables

**Figure 1 antibiotics-12-00393-f001:**
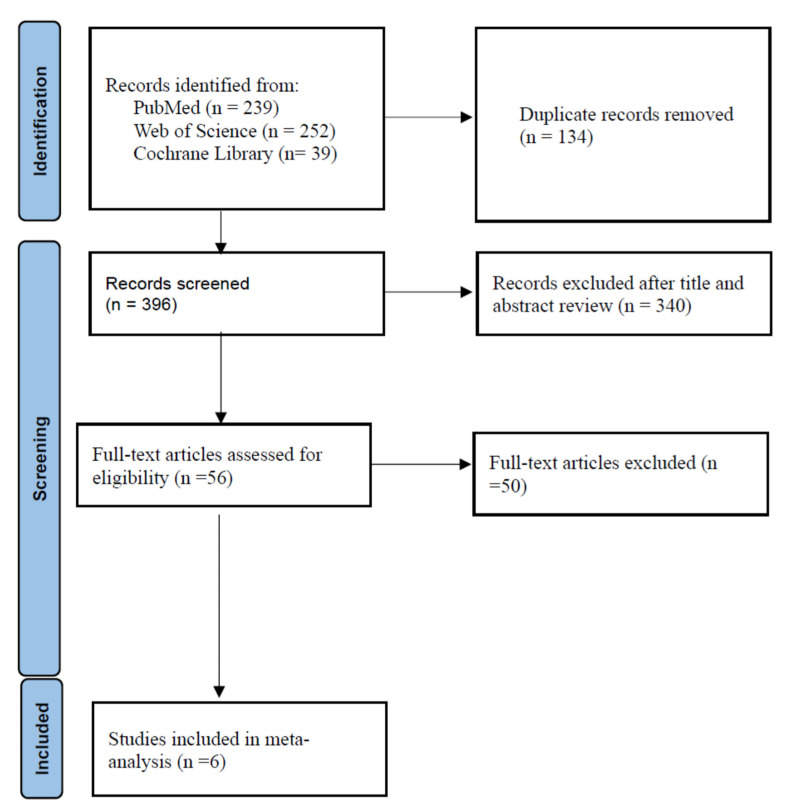
Flow diagram of the study selection process. Six studies were included in the meta-analysis, including one randomized controlled trial, five retrospective observational studies.

**Figure 2 antibiotics-12-00393-f002:**
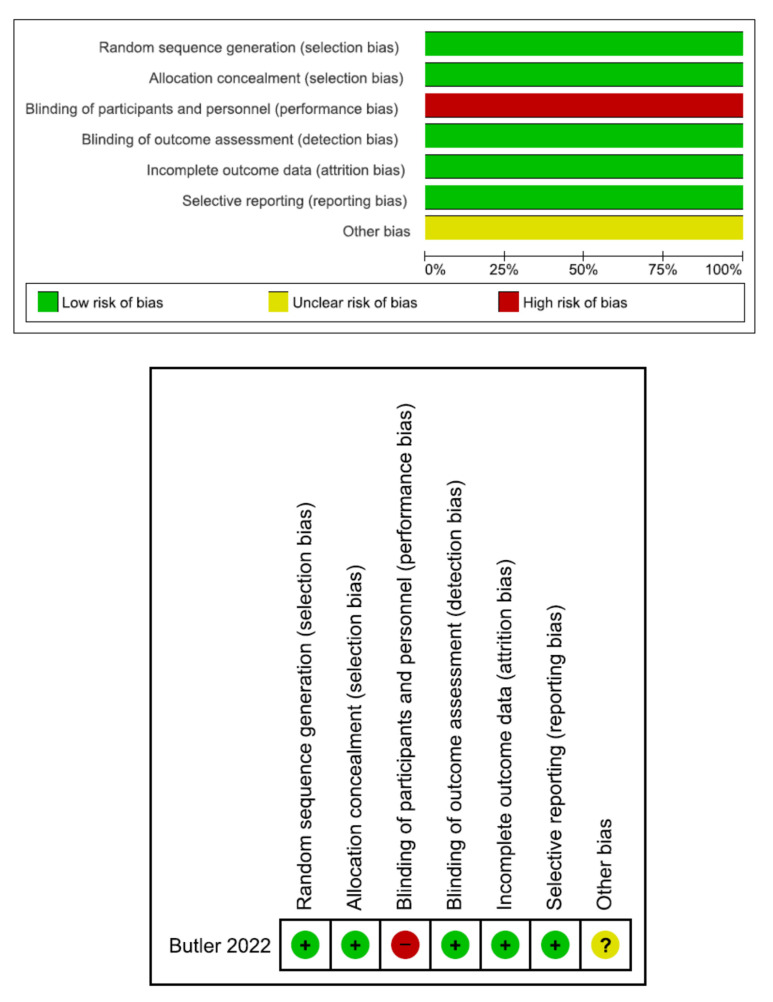
Risk of bias of one randomized controlled trial [26].

**Figure 3 antibiotics-12-00393-f003:**
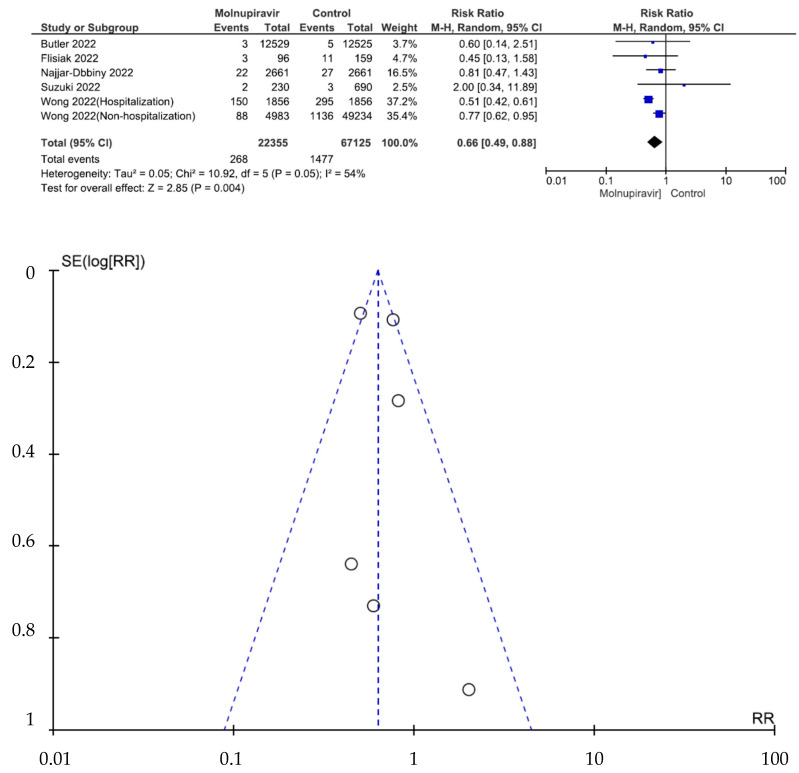
All-cause mortality between molnupiravir and placebo in the treatment of mild to moderate COVID-19 in adult patients [26,27,28,29,30,31].

**Figure 4 antibiotics-12-00393-f004:**
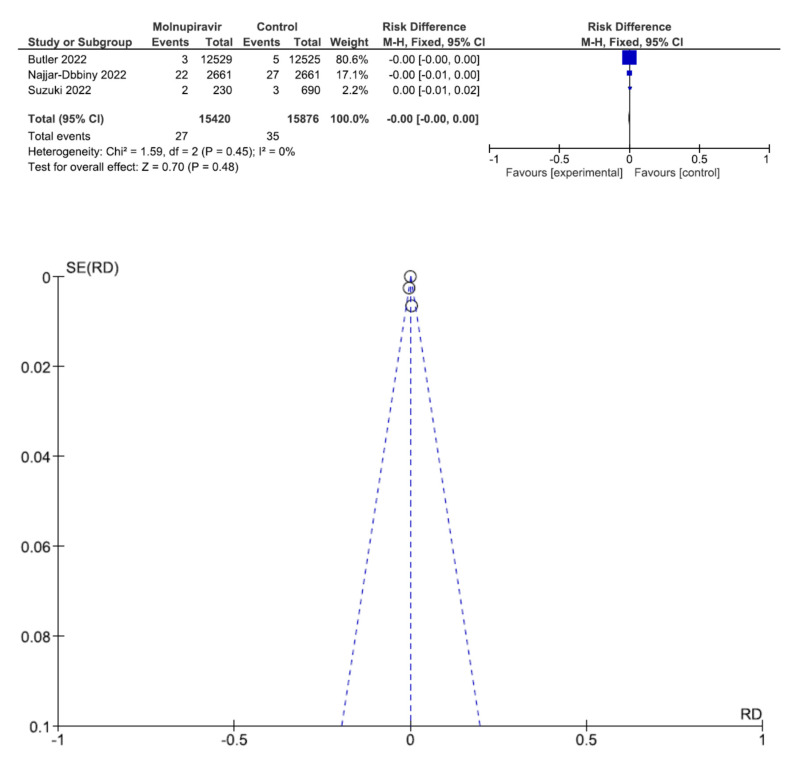
All-cause mortality between molnupiravir and placebo in the treatment of mild to moderate COVID-19 in adult patients with high COVID-19 vaccination coverage [26,28,29].

**Figure 5 antibiotics-12-00393-f005:**
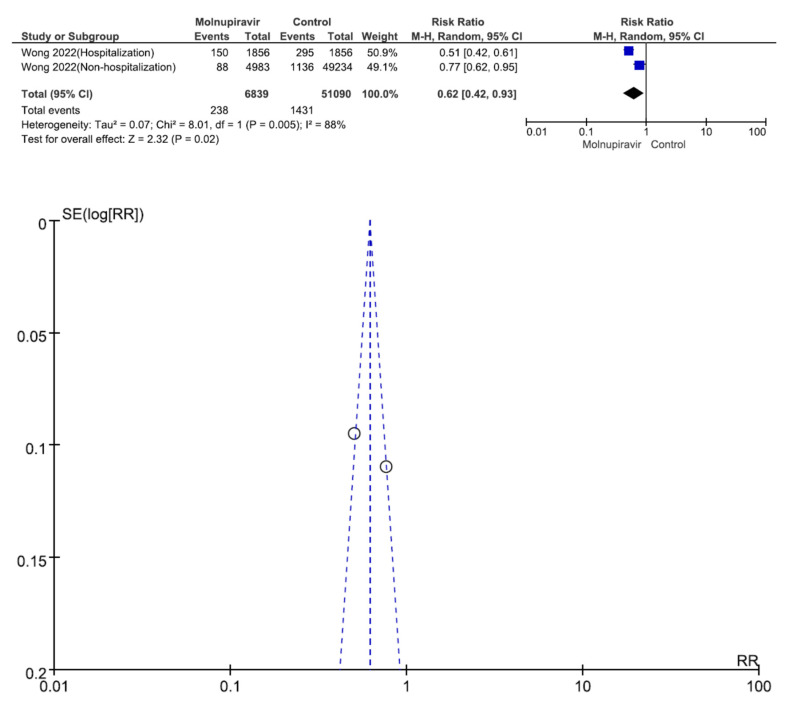
All-cause mortality between molnupiravir and placebo in the treatment of mild to moderate COVID-19 in adult patients with low COVID-19 vaccination coverage [30,31].

**Figure 6 antibiotics-12-00393-f006:**
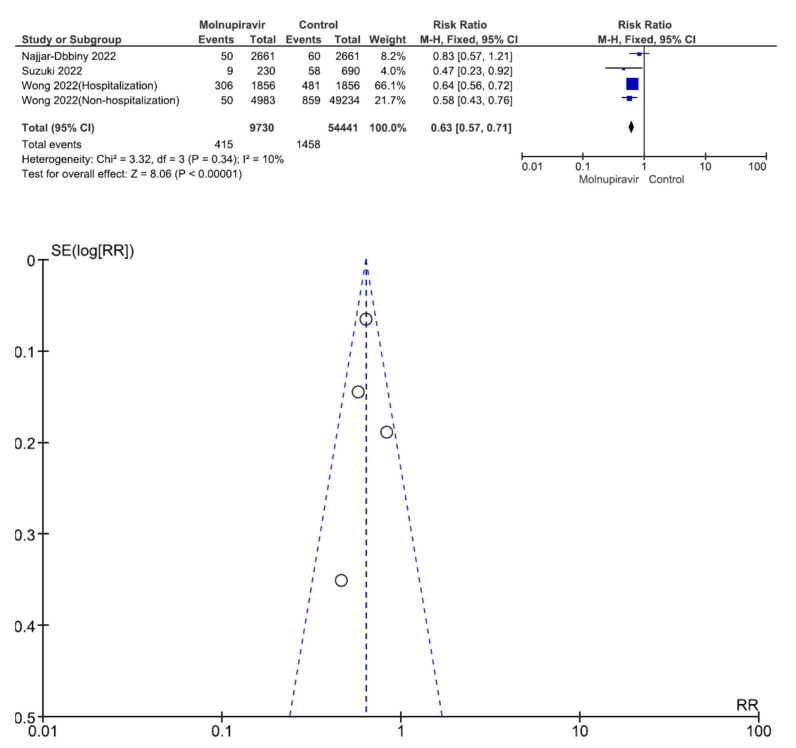
Composite outcome of disease progression between molnupiravir and placebo in the treatment of mild to moderate COVID-19 in adult patients [28,29,30,31].

**Figure 7 antibiotics-12-00393-f007:**
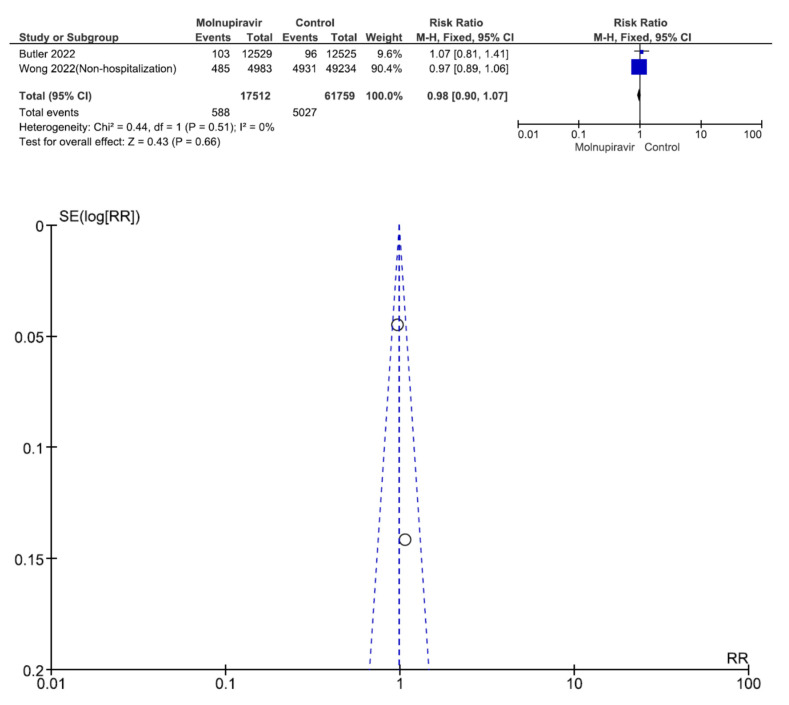
Hospitalization between molnupiravir and placebo in the treatment of mild to moderate COVID-19 in adult patients [26,31].

**Table 1 antibiotics-12-00393-t001:** Characteristics of the included studies.

Author/Study Design	Region	Study Period	Two Vaccine Doses (%)	Setting/SAR-CoV-2 VOC	Mean Age of Patients	Number of Patients
Butler CC, RCT, [26]	UK	December 2021 to April 2022	Mol (96%) Pla (97%)	Non-hospitalization /Omicron	Mol: 56.7 Y/O Pla: 56.5 Y/O	Mol: 12,529 Pla: 12,525
Flisiak R, RET, [27]	Poland	January 2022 to April 2022	No data	Hospitalization /Omicron	Mol: 67.4 Y/O Pla: 67.4 Y/O	Mol: 96 Pla: 159
Najjar-Debbiny R, RET, [28]	Israel	January 2022 to February 2022	Mol (77.3%) Pla (77.3%)	Non-hospitalization /Omicron	Mol: 74.2 Y/O Pla: 55.5 Y/O	Mol: 2661 Pla: 2661
Suzuki Y, RET, [29]	Japan	January 2022 to April 2022	Mol (82.2%) Pla (81.7%)	Hospitalization /Omicron	M: 64.1 Y/O Pla: 64.7 Y/O	Mol: 230 Pla: 690
Wong CK, RET, [30]	Hong Kong	February 2022 to April 2022	Mol (6.2%) Pla (9.0%)	Hospitalization /Omicron BA.2	Mol: 80.8 Y/O Pla: 74.3 Y/O	Mol: 1856 Pla: 1856
Wong CK, RET, [31]	Hong Kong	February 2022 to June 2022	Mol (16.1%) Pla (12.4%)	Non-hospitalization /Omicron BA.2.2	Mol: >60 Y/O (88.7%) Pla: >60 Y/O 92.2%)	Mol: 4983 Pla: 49,234

RCT: randomized controlled trial; RET: retrospective study. Mol: molnupiravir; Pla: placebo; UK: United Kingdom; variants of concern. SAR-CoV-2: severe acute respiratory syndrome coronavirus 2. VOC: variant of concern; Y/O: years old.

**Table 2 antibiotics-12-00393-t002:** Risk bias of five retrospective cohort studies.

Author/Year	Confounding	Selection	Interventions Classification	Interventions Deviations	Missing Data	Measurement of Outcomes	Selective Results
Flisiak R, 2022, [27]	moderate risk	low risk	moderate risk	moderate risk	moderate risk	low risk	moderate risk
Najjar-Debbiny R, 2022, [28]	low risk	low risk	moderate risk	moderate risk	moderate risk	low risk	low risk
Suzuki Y, 2022, [29]	moderate risk	moderate risk	moderate risk	moderate risk	moderate risk	moderate risk	moderate risk
Wong CK, 2022, [30]	moderate risk	low risk	moderate risk	moderate risk	moderate risk	low risk	low risk
Wong CK, 2022, [31]	moderate risk	low risk	moderate risk	moderate risk	moderate risk	low risk	low risk

## Data Availability

The datasets generated during and/or analyzed during the current study are not publicly available but are available from the corresponding author on reasonable request.

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
