# Peer review of "Real-World Clinical Outcomes of Molnupiravir for the Treatment of Mild to Moderate COVID-19 in Adult Patients during the Dominance of the Omicron Variant: A Meta-Analysis"

_antibiotics, 2023, doi:10.3390/antibiotics12020393_

Round 1
Reviewer 1 Report
Conclusively, I suggest accepting this article with a minor revision. This article looks like a review paper. However, I suggest the authors clearly mention their contributions in the conclusion section. The contribution of this manuscript is not apparent. For example, the authors stated in the conclusion section "There was no clinical benefit of reducing all causes of mortality in patients with high COVID-19 vaccination coverage". Did this sentence represent the feature of this paper?
Author Response
Reviewer 1:
Conclusively, I suggest accepting this article with a minor revision. This article looks like a review paper. However, I suggest the authors clearly mention their contributions in the conclusion section. The contribution of this manuscript is not apparent. For example, the authors stated in the conclusion section "There was no clinical benefit of reducing all causes of mortality in patients with high COVID-19 vaccination coverage". Did this sentence represent the feature of this paper?
Reply: In the PANORAMIC study, more than 90% of patients were fully vaccinated, and the study concluded that molnupiravir did not reduce hospitalization or all-cause mortality in COVID-19 patients.
The feature of this paper is” the clinical effectiveness of molnupiravir was associated with COVID-19 vaccination coverage and age in COVID-19 patients. Molnupiravir showed clinical effectiveness in all-cause mortality and composite outcome of disease progression for low COVID-19 vaccination coverage in adult patients with mild to moderate COVID-19 compared with placebo.”
(The manuscript has been carefully reviewed by AJE (American journal Experts).
Reviewer 2 Report
Authors present meta-analysis of clinical reports from 6 studies (after filtering down search criteria).
Well written and executed analysis.
Only suggestion is line 117 "invention" of Omicron is wrong word. Suggest "advent" or "occurence"
Author Response
Reviewer 2:
Authors present meta-analysis of clinical reports from 6 studies (after filtering down search criteria). Well written and executed analysis.
Only suggestion is line 119 "invention" of Omicron is wrong word. Suggest "advent" or "occurence"
Reply: I change to “ In 2021, prior to the advent of Omicron, data from the "MOVe-OUT" experiment were collected in unvaccinated patients.”
(The manuscript has been carefully reviewed by AJE (American journal Experts).
Reviewer 3 Report
Dear authors,
your work is very well detailed and presented.
Results are consistent with goals and methods.
A recent paper has been published on safety and tolerability of Molnupiravir. Please mention and discuss its results.
Mazzitelli et al.
Molnupiravir and Nirmatrelvir/Ritonavir: Tolerability, Safety, and Adherence in a Retrospective Cohort Study.
Regards
Author Response
your work is very well detailed and presented. Results are consistent with goals and methods. A recent paper has been published on safety and tolerability of Molnupiravir. Please mention and discuss its results.
Reply: 1. I cite the reference in my article (in discussion).
In the study by Mazzitelli M et al., 28/407 (6.9%) patients experienced any adverse events following oral molnupiravir: 18 (4.4%) patients who reported adverse events presented grade 1 adverse events, 8 (1.9%) presented grade 2 adverse events and 2 (0.4%) presented grade 3 adverse events. The most reported side effects were nausea/vomiting (2%), bloating (1.7%), and dysgeusia (1.2%). Two patients (0.5%) reported a severe hypersensitivity reaction.
Reviewer 4 Report
Dear Authors,
congratulations on your valuable work. Please, find here below some suggestions that, I hope, could further improve the paper.
1. In both the title and the running title, you mention 'clinical outcome' as singular form. Since you mention more outcomes, it may be more accurate to change the terms to plural.
2. A semantic issue to consider concerns the choice of the word 'efficacy'. You often mention this term, which is typically used in RCTs and in the pre-market authorisation phases of a drug. However, since the evidence comes from real-world studies, perhaps it might be more correct to use the word 'effectiveness'.
3. In the introduction, it could be useful to mention the main differences between molnupiravir and nirmatrelvir/ritonavir, in order to better detail when these two drugs are used and in which categories of patients.
4. Still talking about the two different antivirals, I think it is important to include in the analysis (or at least in the introductory session and discussion), those studies that evaluated the real-world effectiveness and safety of both drugs, in the same setting. Here below you may find some suggested references that, in my opinion, should be added to your paper:
- https://doi.org/10.3390/v15020384
- https://doi.org/10.3390/vaccines10101731
- https://doi.org/10.3390/v15010192
5. Regarding the 'Omicron era', I think it would be better to identify the individual subvariants of interest (VOCs) involved in the period and to specify any changes in sensitivity of the various drugs depending on the different VOC. Omicron, infact, became the dominant circulating VOC since the end of 2021. Please, specify better the time interval you chose and the different circulating sub-variants (maybe, you can add a figure or table about this, which can easily help readers).
6. PRISMA ref is missing, please add it.
7. Regarding vaccination status, you stated that 'High COVID-19 vaccination coverage was defined as more than 70% fully vaccinated patients. Low COVID-19 vaccination coverage defined fully vaccinated patients as less than 71%'. Why did you choose this cut-off? Is there any literature that support this choice?
8. Could you consider a substantial improvement and implementation of the 'Quality assessment and statistical analysis part'? I do not think it is appropriate to cite other studies you have conducted and not cite the methodology you used itself. I would ask you to carefully review this part.
9. Figure 1: you left the formatting symbols. Please, revise the figure.
10. Please, update your discussion according with the recently published refs already mentioned at point n.4
Author Response
Reviewer 4:
congratulations on your valuable work. Please, find here below some suggestions that, I hope, could further improve the paper.
- In both the title and the running title, you mention 'clinical outcome' as singular form. Since you mention more outcomes, it may be more accurate to change the terms to plural.
Reply: I change to “ clinical outcomes”.
- A semantic issue to consider concerns the choice of the word 'efficacy'. You often mention this term, which is typically used in RCTs and in the pre-market authorisation phases of a drug. However, since the evidence comes from real-world studies, perhaps it might be more correct to use the word 'effectiveness'.
Reply: I change to “effectiveness”
- In the introduction, it could be useful to mention the main differences between molnupiravir and nirmatrelvir/ritonavir, in order to better detail when these two drugs are used and in which categories of patients.
Reply:
1.Molnupiravir and nirmatrelvir plus ritonavir have received emergency use authorization from the Food and Drug Administration (FDA) of the United States for the treatment of mild to moderate COVID-19 in adult patients who are not hospitalized and have a high risk of developing severe disease. Nirmatrelvir 300 mg and ritonavir 100 mg are the suggested doses for this medication. Based on the results of the EPIC-HR study, it was found that nirmatrelvir and ritonavir significantly decreased the risk of hospitalization or death due to COVID-19 by 89% among unvaccinated patients within 3 days of symptom onset and by 88% among unvaccinated patients within 5 days of symptom onset.
2.Molnupiravir is a therapy that can be used instead of Paxlovid when this is not available, not possible to use, or not clinically appropriate
- Still talking about the two different antivirals, I think it is important to include in the analysis (or at least in the introductory session and discussion), those studies that evaluated the real-world effectiveness and safety of both drugs, in the same setting. Here below you may find some suggested references that, in my opinion, should be added to your paper:
- https://doi.org/10.3390/v15020384
Reply: (Discussion)
In the study by Mazzitelli M et al., 28/407 (6.9%) patients experienced any adverse events following oral molnupiravir: 18 (4.4%) patients who reported adverse events presented grade 1 adverse events, 8 (1.9%) presented grade 2 adverse events and 2 (0.4%) presented grade 3 adverse events. The most reported side effects were nausea/vomiting (2%), bloating (1.7%), and dysgeusia (1.2%). Two patients (0.5%) reported a severe hypersensitivity reaction. https://doi.org/10.3390/vaccines10101731
- https://doi: 10.3390/vaccines10101731.
Reply: (discussion)
Gentile et al. found that 56.8% of 257 patients received molnupiravir, whereas 43.2% received nirmatrelvir/ritonavir. There were three hospitalizations in the molnupiravir (2.1%) group and one in the nirmatrelvir/ritonavir (0.9%) group. One patient who received molnupiravir died. The authors concluded that in a cohort of mostly vaccinated individuals treated with oral antivirals, hospitalization and death rates were low. The study results were not surprising in either group, as 247 (96.1%) individuals had received at least two doses of a COVID-19 vaccine. In fact, SARS-CoV-2 vaccination likely influenced the overall outcome observed in the study population.
3.https://doi.org/10.3390/v15010192
Reply: (introduction): Minotti et al. reported the clinical off-label prescription experience of two children with COVID-19 who received Molnupiravir.
- Regarding the 'Omicron era', I think it would be better to identify the individual subvariants of interest (VOCs) involved in the period and to specify any changes in sensitivity of the various drugs depending on the different VOC. Omicron, infact, became the dominant circulating VOC since the end of 2021. Please, specify better the time interval you chose and the different circulating sub-variants (maybe, you can add a figure or table about this, which can easily help readers).
Reply: (in new Table 1)
- Butler cc: study period: December 8, 2021 to April 27, 2022 (VOC: Omicron)
- Flisiak R: study period: January 1 , 2022 to April 30, 2022 (VOC:Omicron)
- Najjar-Debbiny R: study period: January 1, 2022 to February 28, 2022 (VOC: Omicron)
- Suzuki Y: study period: January 1, 2022 to April 30, 2022 (VOC: Omicron)
- Wong CK: study period: February 26, 2022 to April 26, 2022 (VOC: Omicron BA.2)
- Wong CK: study period:February 26 and June 26, 2022 (VOC: Omicron BA 2.2)
- PRISMA ref is missing, please add it.
Reply: I add a new PRISMA in supplementary table
- Regarding vaccination status, you stated that 'High COVID-19 vaccination coverage was defined as more than 70% fully vaccinated patients. Low COVID-19 vaccination coverage defined fully vaccinated patients as less than 71%'. Why did you choose this cut-off? Is there any literature that support this choice?
Reply: Two references as below:
- 69.4% of the world population has received at least one dose of a COVID-19 vaccine. 13.27 billion doses have been administered globally, and 1.39 million are now administered each day. 26.4% of people in low-income countries have received at least one dose. (https://ourworldindata.org/covid-vaccinations)
2.Partial vaccination coverage was calculated on the basis of the number of people who uptake at least the first dose of the SARS‐CoV‐2 vaccine. Fully vaccination coverage arrived from the number of people who received all doses prescribed by the initial vaccination protocol, divided by the total population of the country. The median vaccination coverage increased to 76.8% (IQR: 66.6%–81.6%) and 71.9% (IQR: 59.8%–79.1%) for people with partial and full vaccination during the Omicron period, respectively.(J Med Virol. 2023 Jan;95(1):e28118. doi: 10.1002/jmv.28118.) Therefore, we choose that 70% is the cut-off.
- Could you consider a substantial improvement and implementation of the 'Quality assessment and statistical analysis part'? I do not think it is appropriate to cite other studies you have conducted and not cite the methodology you used itself. I would ask you to carefully review this part.
Reply:
Quality assessment
We assessed the risk of bias in each study using the Cochrane Risk-of-Bias Tool 2.0 for RCTs. The Risk of Bias in Non-randomized Studies of Interventions (ROBINS-I) tool was used to evaluate observational studies. We conducted a sensitivity analysis by systematically removing each study and assess the impact of the study quality on the effect estimates. Two reviewers examined publications independently to avoid bias. When disagreement occurred, a third author resolved the issue.
Statistical analysis
Data were entered into the Cochrane Review Manager software RevMan 5. Differences were expressed as odds ratios (ORs) with 95% confidence intervals (CIs) for dichotomous outcomes. The significance of the pooled ratios was determined by the Z test and a P value less than 0.05 was considered statistically significant. The I2 test was used to assess the proportion of statistical heterogeneity, and the Q-statistic test was used to define the degree of heterogeneity. A P value less than 0.10 for the Q-test and I2 less than 50% were considered significant among the studies. The fixed-effects model was used when the effects were assumed to be homogenous, while the random-effects model was used when they were heterogeneous. Publication bias was assessed by examining the funnel plot.
- Figure 1: you left the formatting symbols. Please, revise the figure.
Reply: I correct figure 1.
- Please, update your discussion according with the recently published refs already mentioned at point n.4
Reply: I update the discussion according with the recently published refs already mentioned at point n.4.
(The manuscript has been carefully reviewed by AJE (American journal Experts).

Round 2
Reviewer 4 Report
No further reccomendation needed. I thank the authors for addressing all the suggestions I reported.